# Current Advances in Regenerative Strategies for Dry Eye Diseases: A Comprehensive Review

**DOI:** 10.3390/bioengineering11010039

**Published:** 2023-12-29

**Authors:** Basanta Bhujel, Se-Heon Oh, Chang-Min Kim, Ye-Ji Yoon, Ho-Seok Chung, Eun-Ah Ye, Hun Lee, Jae-Yong Kim

**Affiliations:** Department of Ophthalmology, Asan Medical Center, University of Ulsan College of Medicine, Seoul 05505, Republic of Korea; basantabhujel86@gmail.com (B.B.); 72004218osh@gmail.com (S.-H.O.); kcm8821@naver.com (C.-M.K.); yejii0849@gmail.com (Y.-J.Y.); chunghoseok@gmail.com (H.-S.C.); yeyeyeyeye@gmail.com (E.-A.Y.); yhun777@hanmail.net (H.L.)

**Keywords:** dry eye, dry eye diseases, regenerative medicine, biomaterials, cell therapy

## Abstract

Dry eye disease (DED) is an emerging health issue affecting millions of individuals annually. Ocular surface disorders, such as DED, are characterized by inflammation triggered by various factors. This condition can lead to tear deficiencies, resulting in the desiccation of the ocular surface, corneal ulceration/perforation, increased susceptibility to infections, and a higher risk of severe visual impairment and blindness. Currently, the clinical management of DED primarily relies on supportive and palliative measures, including the frequent and lifelong use of different lubricating agents. While some advancements like punctal plugs, non-steroidal anti-inflammatory drugs, and salivary gland autografts have been attempted, they have shown limited effectiveness. Recently, there have been promising developments in the treatment of DED, including biomaterials such as nano-systems, hydrogels, and contact lenses for drug delivery, cell-based therapies, biological approaches, and tissue-based regenerative therapy. This article specifically explores the different strategies reported so far for treating DED. The aim is to discuss their potential as long-term cures for DED while also considering the factors that limit their feasibility and effectiveness. These advancements offer hope for more effective and sustainable treatment options in the future.

## 1. Introduction

Dry eye disease (DED) is a highly prevalent ocular surface condition, impacting roughly 5–50% of adults globally [1]. A recent international Dry Eye Workshop (DEWS II) redefined DED as “a multifactorial condition of the eye surface marked by an imbalance in the tear film, accompanied by eye-related symptoms.” “This imbalance involves tear film instability and hyperosmolarity, inflammation and damage to the eye surface, and abnormalities in the neuro-sensory functions” [2]. In 2017, the Tear Film and Ocular Surface Society International Dry Eye Workshop II (TFOS DEWS II) issued a report that classified DED into two primary types: aqueous tear-deficient DED and evaporative DED. The former can be further categorized into two subtypes: Sjögren syndrome (SS)-related DED and non-SS-related DED [3].

While DED does not lead to blindness, its discomfort significantly reduces the quality of life, hindering both physical and social aspects for affected individuals [4]. The primary treatment for relieving symptoms at all levels of DED is the use of artificial lubricants [5]. Yet, the prolonged use of artificial lubricants with preservatives can potentially harm the ocular surface epithelium [6]. In an effort to prevent side effects triggered by preservatives, alternative preservative-free versions have been designed. Still, their utilization of disposable containers increases costs, and the requirement for frequent application hampers their broad acceptance.

For moderate to severe DED, topical corticosteroids help reduce inflammation, but their prolonged use increases the risks of glaucoma and cataracts [7]. Conversely, cyclosporin ophthalmic emulsions are recommended as a long-term anti-inflammatory treatment for patients with moderate to severe DED [8]. Nevertheless, patients’ adherence to cyclosporine ophthalmic emulsions is affected by side effects like ocular burning and stinging sensations [9]. In addition, conventional topical ophthalmic solutions face a major drawback in their restricted drug presence on the eye’s surface, which is due to nasolacrimal drainage, blinking, drug dilution by tears, reflex tearing, non-therapeutic absorption by the cornea, patient non-compliance, and metabolic degradation [10]. To counter their diminished ocular bioavailability, it is recommended to administer frequent, high-concentration doses. However, this approach leads to heightened side effects [11].

Despite some advancements, the understanding and treatment of DED still face significant challenges. At present, dry eye has no definitive cure, and the available treatments primarily focus on alleviating the symptoms of DED. The main goal is to interrupt the vicious cycle of DED, prevent the condition from becoming chronic, and halt its progression [2]. To tackle the complexity of this condition and devise more effective diagnostic and therapeutic approaches, researchers have been increasingly adopting interdisciplinary methods. Hence, researchers have investigated different approaches in regenerative medicine, pharmacology, nanotechnology, and biomedical engineering to address these challenges. These efforts involve the development of innovative nano-systems, hydrogels, and contact lenses aimed at enhancing the administration of drugs to the surface of the eye. Additionally, the utilization of mesenchymal stem cells (MSC) and the paracrine and autocrine effects of the secretome derived from these cells have been explored for their ability to modulate the immune response and encourage the regeneration of the eye’s epithelial tissue.

In this review, we will explore the advancements made in DED treatment, focusing on emerging ideas and interdisciplinary approaches such as regenerative medicine, cell-based therapies, and tissue engineering that hold great potential to shape the future of the field (Figure 1). For this review, we conducted a comprehensive search on PubMed (https://pubmed.ncbi.nlm.nih.gov/ (accessed on 1 October 2023) and Google Scholar using the terms “Dry eye diseases”, “Regenerative therapies”, and “Biomaterials” as keywords from 1999 to April 2023.

## 2. Pathology of DED

The pathogenesis of dry eye is complex and not fully understood. Current research on the dry eye mechanism predominantly concentrates on the following areas:

### 2.1. Inflammation in Dry Eye

Dry eye is associated with inflammation, and measurements of tear cytokine levels in affected individuals indicate ocular surface inflammation [12]. Studies have shown elevated levels of various cytokines, including interleukins (ILs)-1, 4, 8, 10, 17A, and 6; matrix metalloproteinase (MMPs)-3 and 9; tumor necrosis factor-beta (TNF-β); and tumor necrosis factor-alpha (TNF-α) in dry eye patients, and their levels correlate with disease severity [13,14,15]. IL-6 plays a significant role by activating inflammatory pathways and inducing the release of IL-17 and other inflammatory factors, leading to increased inflammation and cell apoptosis [16]. IL-17 further encourages the secretion of MMP-3 and MMP-9, which contribute to corneal barrier breakdown during dry eye development [17].

In 2017, Rhee and Mah described a vicious cycle of dry eye: diverse factors disrupt the tear film instability, leading to tear hyperosmolarity (Figure 2A), neurogenic inflammation, corneal and conjunctival apoptosis, and meibomian and lacrimal glands activation, worsening the instability of the tear film. Consequently, anti-inflammatory treatment is crucial for addressing DED [18].

### 2.2. Immune Response of Dry Eye

The innate immune response at the ocular surface initiates acute inflammation through the upregulation of mitogen-activated protein kinase (MAP-K), Jun N-terminal kinase (JNK), extracellular regulated protein kinase (ERK), and p38, stimulating nuclear transcription factor-κB (NF-κB). This leads to the release of pro-inflammatory mediators, such as TNF-α, IL-1, IL-6, *chemokine (C-C motif) ligand 3* (CCL3), *chemokine (C-C motif) ligand* 5 (CCL5), and *chemokine (C-C motif) ligand 20* (CCL20) [19]. 

IL-1 and TNF-α activate antigen-presenting cells (APCs), which adhere to lymphatic endothelial cells via upregulated intercellular adhesion molecule-1 (ICAM-1). APCs then migrate to regional lymph nodes in a chemokine receptor 7 (CCR7)-mediated manner [20]. The adaptive immune response involves APCs engaging with Th0 cells in lymph nodes, leading to the differentiation of Th0 cells into subsets (Th1, Th2, Th17, and Tregs) based on cytokine signaling factors [21]. T cell differentiation relies on the equilibrium of signaling molecules expressed by mature APCs (mAPCs), including transforming growth factor-beta (TGF-β), IL-6, IL-12, IL-17, IL-23, and interferon-gamma (IFN-γ) [22]. The balance of these cytokines influences the specific subtype differentiation of effector T cells.

Effector T cells then travel through the efferent arm of blood vessels to the conjunctival stroma, where they reactivate resident mAPCs and are recruited to the ocular surface [23,24]. Specifically, Th1 and Th17 cells play a crucial role in causing ocular surface damage and inflammation associated with DED by releasing cytokines that disrupt normal balance and tear dysfunction [25]. This triggers an immune response, perpetuating the cycle of DED pathogenesis (Figure 2B). In nonpathogenic responses, Tregs are responsible for suppressing the effector response and regulating immunity [26,27].

### 2.3. Other Pathologies of Dry Eye

Any interruption in the nerve signal transmission pathway can result in DED. The cornea contains many sensory nerves that detect alterations of the eye’s surface, such as changes in tear composition and inflammation. These nerves send signals to the nerve center in response. The nerve center then stimulates gland secretion and blinking, essential for tear film distribution across the whole ocular surface [28]. Atypical corneal nerve function worsens ocular surface damage, prolonging dry eye-related inflammation. Additionally, conjunctival goblet cells’ secretions are affected by parasympathetic or peripheral nerves. This emphasizes the significance of normal conjunctival nerve function in maintaining ocular surface health. Prior studies have demonstrated that oxidative stress can damage the ocular surface in dry eyes. Tear film components, including 8-hydroxy-2′-deoxyguanosine (8-OHdG) and malondialdehyde (MDD), vary seasonally, highlighting their association with this condition.

**Figure 2 bioengineering-11-00039-f002:**
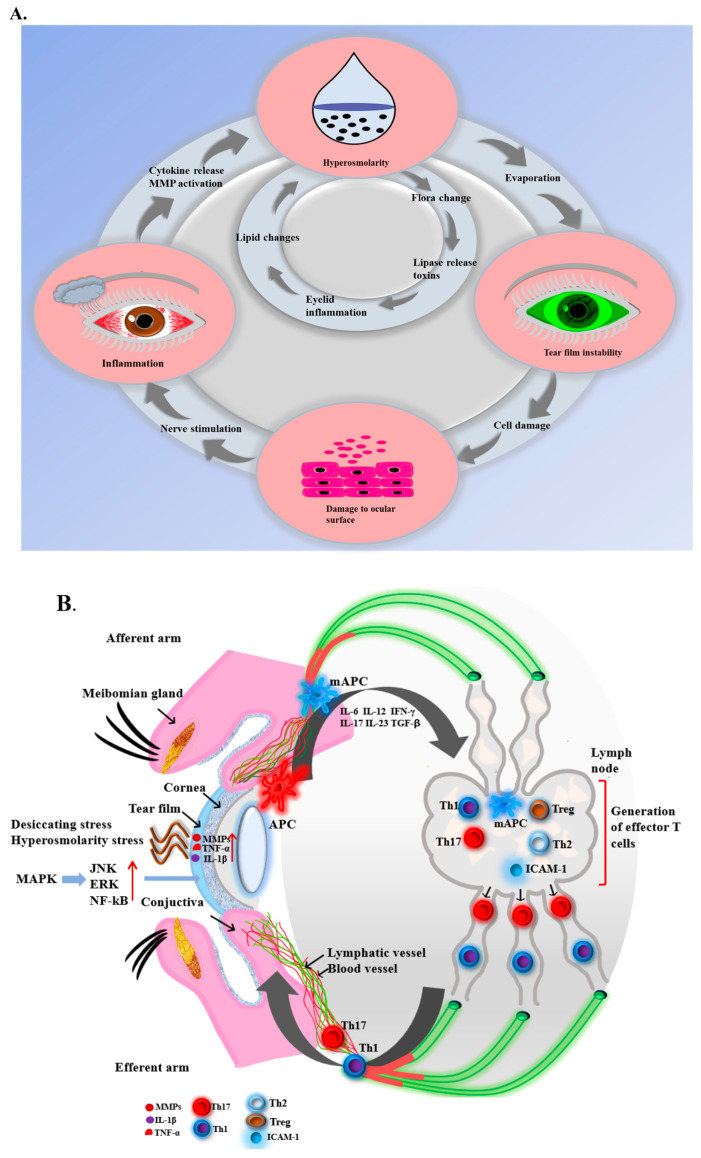
(**A**) Vicious Cycle in Dry Eye Pathology. A detrimental loop in DED can be initiated by a range of factors. Prolonged exposure to conditions that harm eye cells induces inflammation, resulting in the release of cytokines and matrix metalloproteinases (MMPs) onto the surface of the eye. This inflammation destabilizes the tear film, causing hyperosmolarity of the tears and perpetuating the vicious cycle [29]. (**B**) DED triggers an immunoinflammatory response on the ocular surface. Desiccation, oxidative factors, and hyperosmolarity stress activate signaling pathways, leading to the release of pro-inflammatory cytokines and facilitating the maturation of APCs. In lymph nodes, mature APCs stimulate effector T cells (Th1 and Th17) to migrate back to the ocular surface. This triggers an immune response causing ocular surface damage and perpetuating the cycle of DED pathogenesis by releasing inflammatory cytokines [30,31].

## 3. Treatments and Weaknesses of Existing Treatments for DED

DED treatment follows a tiered approach based on disease severity and factors like subclinical inflammation of the ocular surface, meibomian gland dysfunction, and underlying systemic diseases. Avoiding risk factors such as cigarette smoking, air conditioning, and dry heating air is crucial [32]. Presently, several medications (including Lacritin, Lubricin, Rapamycin, Estradiol, Thymosin, Ecabet sodium, Visomitin, and others) are undergoing clinical trials as treatments for DED [33].

DED medications are mainly available in the form of eye drops or emulsions. Upon application, these substances enter the nasolacrimal duct but are rapidly cleared through lymphatic flow and blood vessels in the conjunctiva. Consequently, only a minimal 1–5% of the drug is absorbed by the intended target tissue. Lipophilic drugs have a bioavailability of less than 5%, while hydrophilic drugs have a bioavailability of only 0.5%. To compensate for this, regular administration of potent medications in high concentrations is essential, but this often results in low patient adherence, especially for long-term eye conditions such as dry eye syndrome (DES). Moreover, a significant portion of the administered drug crosses into the systemic circulation, potentially causing side effects in major organs due to escaping first-pass metabolism [34]. Between 18.2% and 80% of cases may experience bacterial contamination when using eye drops due to contact of contaminated surfaces with the dispenser tip. Maintaining a consistent delivery of the exact number of drops is difficult, with around 11.3–60.6% of patients finding it challenging to achieve. Additionally, the quantity of drug emitted from the eye drop bottle varies depending on the force applied, leading to dosage inconsistencies. As a result, even with careful handling, conventional eye drops may not deliver the correct drug amount, leading to patient dissatisfaction and poor clinical outcomes [35].

## 4. Biological Approaches in DED

### 4.1. Autologous Serum (AS)

Several studies have demonstrated the effectiveness of AS for DED [36,37,38,39]. DED is a persistent and sight-threatening condition resulting from inadequate lacrimal gland function, frequently linked to immune-related disorders such as Sjogren’s syndrome. The primary treatment for DED is the use of artificial tear eye drops to supplement the lack of tears. Artificial tears offer limited relief in severe cases of DED. For these patients, AS eye drops, created from the patient’s blood, prove beneficial because they contain crucial growth factors absent in artificial tears. These serum eye drops have been employed for years to address DED and persistent corneal epithelial defects, yielding positive subjective and objective outcomes. Nevertheless, the use of AS eye drops comes with its own set of limitations and difficulties.

The justification for using AS stemmed from the observation that AS possesses numerous elements found in tears, suggesting its potential to function as an alternative to natural tears for managing dry eye symptoms. Tsubota et al. studied AS eye drops for Sjögren-type dry eyes and found an increased goblet cell count, reduced squamous metaplasia, and enhanced MUC-1 expression in conjunctival epithelial cells [40]. Ogawa et al. found that AS eye drops were effective and safe in treating dry eye in chronic graft-versus-host disease (GVDH), which typically does not respond to conventional therapies. The cornea, which is highly innervated by trigeminal nerves, is a very sensitive tissue. Reduced corneal sensitivity can lead to the development of neurotrophic ulcers [41]. In a study by Noda-Tsuruya et al., it was reported that the use of AS eye drops led to an improvement in vital staining following laser-assisted in situ keratomileusis (LASIK) [42]. Further, AS eye drops were found to be effective and safe for dogs with refractory keratoconjunctivitis sicca by improving tear film stability, ocular surface health, and overall clinical symptoms [43].

The mode of action of AS is to mimic the biochemical characteristics of natural basal tears, with the aim of healing the ocular surface epithelium [44]. AS closely mimics the natural composition of unstimulated human tears, matching their pH (7.4) and osmolality (296-8 mOsm/kg H_2_O). Additionally, AS contains similar or elevated amounts of growth factors like epidermal growth factor (EGF), transforming growth factor-β (TGF-β), lysozyme, vitamin A, and fibronectin when compared to natural tears [36]. These epitheliotropic factors are believed to contribute to the therapeutic effects of AS both in vitro and in vivo [45,46].

### 4.2. Allogenic Serum (ALS)

ALS from healthy blood donors has been employed in cases where the patient’s own serum is inappropriate or inaccessible, or when frequent blood sampling is impractical. This includes patients with viral infections, severe anemia, septicemia, and elderly individuals with multiple systemic diseases [47,48]. ALS eye drop usage provides an extra benefit in immune-mediated conditions with extensive inflammation and systemic involvement. It helps avoid the direct application of AS with high levels of pro-inflammatory cytokines to the eye [49]. In addition, ALS enables the generation of ample tears, enhances treatment logistics, and facilitates the standardization and screening of the cytokine composition, anti-inflammatory agents, and epitheliotropic components. This improves the overall efficacy of the treatment [47].

ALS contains a substantial amount of albumin, a protein recognized for its antioxidant properties. More precisely, the free thiol group on cysteine 34 of albumin enables it to readily engage with and counteract free radicals of oxygen and nitrogen. This capability could provide safeguarding for the ocular surface against oxidative damage [50].

### 4.3. Platelet-Rich Plasma (PRP)

PRP refers to various blood-derived products obtained by centrifugation that contain higher platelet concentrations than circulating blood. When artificially activated, the platelets release a pool of proteins and factors, including EGF, TGF-β, platelet-derived growth factor (PDGF), vascular endothelial growth factor (VEGF), insulin-like growth factor-1 (IGF-1), hepatocyte growth factor (HGF), nerve growth factor (NGF), and platelet factor-4 (PF-4), crucial for corneal and conjunctival wound healing [51,52]. Various blood-derived products with different compositions, platelet concentrations, and leukocyte presence or absence can be obtained, including PDGF, plasma rich in growth factors (PRGF), PC, LR-PRP, LP-PRP, and more [53,54]. Over 40 preparation methods exist for PRGF, hindering comparisons of efficacy and safety in scientific studies [51,55]. Industrialized PRGF eye drops have shown preserved biological activity for 3 months in human use, benefiting severe dry eye patients unresponsive to conventional therapy, particularly in neutrophic cases. Additionally, PRGF may aid in the growth of limbal stem cells [56,57].

The cellular composition of PRP determines the levels of growth factors and catabolic cytokines. A higher platelet concentration enhances anabolic signaling through increased growth factor levels [53]. Leukocytes, on the other hand, influence PRP quality by elevating catabolic signaling molecules like MMP-9. The leukocyte concentration is directly correlated with PDGF and VEGF levels but negatively correlated with fibroblast growth factor (FGF) levels [54]. Non-platelet components of whole blood contribute to its biological effects. Red and white blood cells can trigger undesirable inflammatory reactions. While leukocytes may have a negative pro-inflammatory effect, some studies indicate that non-platelet cellular components are crucial for optimal platelet function, including coagulation, growth factor release, and serum’s ability to stimulate cell proliferation [53,55].

Several studies have examined the safety and effectiveness of diverse PRP formulations for treating dry eyes. These studies have demonstrated enhanced tear film quality and reduced symptom severity, even in individuals who had received prior treatments with artificial tears [58]. Moreover, an uncontrolled clinical study involving 368 patients revealed that applying autologous PRP topically effectively reduced dry eye symptoms in 322 of these patients. Platelets secrete growth factors that aid in reconstructing the ocular surface, unlike AS [59]. However, PRP comes with restrictions on its usage, such as avoiding its use in cases of severe heart conditions, active bacterial infections, and a medical history involving specific viral diseases like hepatitis B and human immunodeficiency virus (HIV) [51].

### 4.4. Umbilical Cord Blood Serum (UCS)

UCS, similar to peripheral blood serum, contains abundant tear-related components. Relative to blood serum, UCS contains three times the amount of EGF and double the quantity of TGF-β. It also features increased levels of NGF and SP but lower concentrations of IGF-1 and vitamin A, although these levels still surpass those found in normal tears [60]. UCS exhibits a bacteriostatic effect due to the presence of antibacterial agents like lysozyme, immunoglobulin G, and complement [58]. UCS eye drops, prescribed at a 20% concentration, are commonly administered 4 to 6 times a day and can be stored at −20 °C for 3 to 6 months [60]. They effectively treat several ocular surface diseases, such as severe dry eye, GVDH complications, neurotrophic keratopathy, and complications after corneal refractive surgery [60,61].

The mechanism of action of UCS is likely to be the same as that of AS, with the distinction lying in a higher concentration of growth factors, which may in fact stimulate the growth of stem cells and thus lead to faster healing [61]. UCS eye drops surpass AS eye drops in effectiveness, demonstrating superior results in alleviating symptoms and enhancing epitheliotropic effects. They achieve this by increasing the density of goblet cells in individuals with severe DES [62]. A significant sample can be collected from the umbilical vein during delivery, meeting the needs of multiple patients simultaneously. Nevertheless, it is essential to evaluate the possibility of allergic reactions and the potential spread of infectious diseases [61]. While further detailed research and understanding of the molecular mechanisms are necessary to confirm their clinical usefulness, all of these studies demonstrated a straightforward and flexible technology that could be explored for regenerative medicine in DED.

### 4.5. Gene Therapy

One of the promising approaches for treating DED is gene therapy. Efficiently delivering target molecules (DNA, mRNA, miRNA, siRNA, and antisense oligonucleotides) using viral and non-viral methods is crucial for achieving therapeutic benefits. However, viral strategies, while effective and specific, pose a risk of viral infection. On the other hand, non-viral vectors, although safer, currently suffer from low delivery efficiency, presenting a significant obstacle in clinical applications [63]. Various approaches, including topical drops as well as intrastromal and subconjunctival injections, can be explored to enhance the accurate and targeted delivery of the therapeutic gene [64]. Gene therapy for treating inherited corneal dystrophies operates through three methods: (a) suppressing or deactivating the mutated gene causing harmful effects on cells; (b) repairing or substituting the mutated gene with a functional version of a healthy gene; or (c) introducing a healthy version of the gene that will produce the therapeutic protein elsewhere to alleviate the disease symptoms [64].

Recent studies in vitro and in vivo have shown promising results for developing anti-inflammatory treatments for lacrimal glands. For instance, in a rabbit model with autoimmune tendencies, the introduction of a suppressor gene to inhibit TGF-β resulted in enhanced tear production, improved stability of the tear film (as indicated by tear film breakup time (TBUT) and rose bengal scores), and decreased infiltration of lymphocytes, specifically T cells and CD18+ cells. This intervention revitalized tear production and alleviated corneal defects [65]. Thomas and colleagues achieved comparable outcomes by utilizing adeno-associated virus (AAV) vectors to express the viral (v) IL-10 gene in the lacrimal gland of a rabbit model with immunopathology and ocular surface disease [66]. Furthermore, in animal models, the application of AAV vectors for gene therapy targeting the aquaporin-1 gene and MUC5AC gene has demonstrated notable relief of dry eye symptoms [67]. These discoveries indicate the potential of gene therapy for addressing DED in humans. Further clinical trials are essential to validate its effectiveness [68].

### 4.6. Artificial Solutions with/without Anti-Inflammatory Drugs

Artificial solutions, which are convenient and primary for eye diseases, often use hyaluronic acid (HA) as a key ingredient for DED. Studies indicate that 0.3% HA is more effective than lower concentrations, improving tear stability and reducing ocular surface damage [69]. Tacrolimus, a steroid-sparing anti-inflammatory agent, has shown positive outcomes in enhancing tear stability and ocular health. Recent research has combined multiple drugs in these solutions, like epigallocatechin gallate (EGCG) and HA or omega-3 EFA and HA, leading to improved tear secretion, reduced corneal damage, and decreased inflammation and oxidative stress markers on the ocular surface [32,70].

HA disrupts the ongoing cycle of DED, interrupting its self-sustaining physiological processes [71]. Osmotic stress, resulting from increased tear film osmolarity, constitutes a primary factor in the detrimental cycle of DED, leading to inflammation and damage to the ocular surface. HA mitigates these issues by lubricating the ocular surface with its viscous, mucoadhesive, and non-Newtonian properties, reducing shear forces. Additionally, HAs established anti-inflammatory effects contribute to alleviating frictional damage, conjunctival thinning, pain, and inflammation associated with DED [12]. 

### 4.7. Nonsteroidal Anti-Inflammatory Drugs and Antibiotics (NSAIDs)

Eye drops containing NSAIDs are employed to alleviate inflammation linked to DED. This medication is primarily recommended for its anti-inflammatory properties rather than its antibacterial effects. NSAIDs work by suppressing cyclooxygenase activity, thereby inhibiting the synthesis of prostaglandins and reducing the migration and phagocytosis of granulocytes and monocytes, ultimately mitigating inflammation on the ocular surface [72].

Eye drops with medications such as diclofenac sodium and ketorolac help with inflammation in dry eyes. Ointments with antibiotics like erythromycin and bacitracin are used for meibomian gland dysfunction [73]. A liquid solution of tetracycline is created for long-term dry eye treatment, mainly for its anti-inflammatory benefits rather than its antibacterial properties [74,75].

### 4.8. Punctal Bags

A tiny device called a “punctal plug” is placed in the tear duct of the eye to stop tears from draining away, helping with dry eyes. Studies show that using these plugs can alleviate dry eye symptoms. They are typically used for people with more serious dry eye issues, and after insertion, it is still important to use artificial tears. Patients should be educated about their use, and regular check-ups are advised to detect plug loss and ensure the condition is well-managed [76].

### 4.9. Corticosteroids

Topical corticosteroids, such as loteprednol etabonate, dexamethasone, prednisolone, and fluorometholone, are proven to work for inflammation linked to dry eyes. The FDA has already approved them for treating inflammatory conditions of the conjunctiva, cornea, and anterior globe [77,78].

In a study with 41 patients experiencing moderate-to-severe DED, the use of topical 0.1% fluorometholone demonstrated superior improvement in corneal and conjunctival staining, hyperemia, and TBUT compared to a control group receiving polyvinyl alcohol drops. Additionally, corticosteroid therapy was found to be more effective in relieving exacerbated dry eye symptoms following a 2 h exposure to adverse environmental conditions, such as humidity and directed airflow to the eyes [79]. Corticosteroids, particularly when used briefly, may be an option for treating DED; however, potential side effects should be considered [75].

### 4.10. Vitamin A

Vitamin A is vital for healthy eyes and is naturally present in tears. It helps to produce the innermost lubricating layer of tears called the mucin layer. If there is a lack of Vitamin A, it can lead to the loss of this layer and damage to goblet cells [80]. Vitamin A drops protect the eyes from various issues, like free radicals and inflammation. Some studies explore using retinoic acid therapy along with vitamin A to treat dry eyes. Applying effective amounts of vitamin A or retinoids can help address dry eye problems [80].

### 4.11. Omega 3 Fatty Acids

Opthalmologists nowadays recommend the oral formulation of essential fatty acids (EFAs) [81]. EFAs serve as building blocks for hormones that handle inflammation. For people with dry eyes, these fatty acids can help by lowering inflammation and changing the makeup of certain eye lipids [82]. Clinicians might suggest eating foods rich in n-3 fatty acids or taking omega-3 supplements like Thera Tears and Bio Tears to ease dry eye discomfort.

A study at the Massachusetts Eye Research Institute, led by Rashid and his team, showed that applying a specific fatty acid directly to the eyes can help treat dry eye symptoms. The use of topical alpha-linolenic acid (ALA) was found to noticeably reduce dry eye signs and inflammation both at the cellular and molecular levels [83].

Certainly, various commercial products, including Bio Tears and Thera Tears, have underscored the significance of certain fatty acids in addressing DED. Studies suggest that applying α-linolenic acid topically could present a novel approach to treating inflammatory changes and clinical manifestations in keratoconjunctivitis. More precisely, the topical use of α-linolenic acid has been shown to notably diminish both molecular and cellular indicators of inflammation and DED symptoms [83]. Table 1 summarizes the clinical trials using various biological approaches (Autologous serum, Allogenic serum, Platelet-Rich plasma, and Umbilical cord blood serum) in treating DED.

## 5. Cell-Based Regenerative Approaches

Inflammatory conditions like dry eye affecting the eye’s surface often involve heightened inflammation, abnormalities in tear film and epithelium, increased osmolarity, and oxidative stress. Researchers have extensively explored the potential of stem cells derived from various origins, including limbal epithelial stem cells, iPSCs, and MSCs, along with their paracrine functions (Figure 3). iPSCs are created by reprogramming skin or blood cells, returning them to a pluripotent embryonic state. In this state, they can be guided to develop into any desired cell type.

Injury and inflammation are responsible for triggering the mobilization, migration, and colonization of stem cells in damaged areas. Specific signaling molecules attract these cells, prompting their mobilization from the bone marrow into the bloodstream. Circulating stem cells then migrate to the injured cornea, where they adhere and contribute to the healing process. The exact mechanism of this homing process is not fully understood [93].

Many researchers have demonstrated the use of MSCs in different animal models of corneal wound healing and chemical burns, resulting in accelerated healing and a faster regeneration of corneal epithelial tissue [94]. Xu et al. demonstrated that intravenous MSC administration inhibited SS in animal models and patients. T cells targeted Treg and Th2, suppressing Th17 and T follicular helper cell responses, thereby alleviating symptoms [95]. Beyazyıldız et al. demonstrated that applying MSCs topically in a rat model of benzalkonium chloride (BAC)-induced DED was safe and effective. They observed significant improvement in tear volume, ocular surface evaluation, and MSC integration in conjunctival goblet cells and meibomian glands. Possible reasons for the improvements included MSC homing, paracrine effects, differentiation into goblet or glandular cells, and stimulation of repair mechanisms. The microvillus structure of the epithelium remained intact, and no neutrophilic infiltration was observed [96]. Dietrich et al. induced dry eye in mice through duct ligation, resulting in acute aqueous-deficient dry eye (ADDE). They used unique extrinsic MSCs from murine lacrimal gland tissue, which led to accelerated regeneration, increased viable structures, reduced inflammation, and improved tissue health compared to saline injection. This suggests the potential advantages of extrinsic MSCs for severe DED patients with limited intrinsic regenerative capacity in the lacrimal gland [97]. In a clinical trial by Hansen et al., seven Sjogren’s disease patients were treated with allogenic adipose-derived MSCs, and no adverse events were observed when MSCs were injected directly into the lacrimal gland capsule. Significant improvements were seen in various eye parameters, likely due to the MSCs anti-inflammatory activity. This therapy is more effective in early-phase DED patients than in those with cicatricial conjunctivitis [98].

A recent research study explored the promising effect of exosomes, which contain bioactive molecules and microRNA (miR-204) released by MSCs, as a topical remedy for patients with GVHD. These exosomes functioned similarly to the MSCs secretome and demonstrated promising results in improving dry eye symptoms [99]. In an individual with GVHD, miR-204 specifically targeted IL-6R, inhibiting the IL-6/IL-6R/Stat3 pathway and encouraging the transformation of inflammatory M1 macrophages into the immunosuppressive M2 phenotype. This transformation proved advantageous for patients. Shen et al. observed less apoptosis and rapid proliferation in rabbit corneal stromal cells when cultured with rabbit adipose MSC-Exos compared to those without MSC-Exos [100]. Similarly, Wang et al. used mouse AD-MSC-derived exosomes (mADSC-Exos) to treat dry eye in mice. They observed that mADSC-Exos reduced cytokine levels, facilitated corneal epithelial repair, and enhanced tear secretion by suppressing the NLRP3-IL-1β signaling pathway [101]. Earlier experimental investigations have demonstrated that MSCs derived from the cornea can effectively inhibit scar formation following corneal injury in mice. These MSCs also promote the regeneration of hyaline interstitial tissue, and this regenerative capability relies on the delivery of miRNAs through exosomes [102]. Moreover, human cornea-derived MSCs can be internalized by corneal epithelial cells, leading to an acceleration in the healing process of corneal injuries [103,104]. Zhou et al. utilized MSC-Exos to address chronic dry eye linked to GVHD in mice as well as persistent dry eye in patients resistant to treatment. Their study revealed that MSC-Exos effectively alleviated dry eye symptoms, with a significant role attributed to the presence of Mir-204 in the exosomes [99]. However, many obstacles remain, inhibiting the use of these approaches in clinical settings. Moreover, cell-based approaches show promise but need further exploration, and standard protocols should be made mandatory. Long-term solutions require more investigation. Table 2 summarizes the different pre-clinical and clinical studies using cell-based approaches to treating DED.

## 6. Biomaterials for DED Treatment

The history of ophthalmic biomaterials is relatively short. The primary objective of advancing successive generations of biomaterials is to address the shortcomings of previous versions and enhance safety, effectiveness, and comfort. Innovations have been made to elevate quality standards and production efficiency, ultimately reducing costs. Market demands to enhance competitiveness and accessibility have further intensified the pressure to cut expenses. Ophthalmic biomaterials have evolved into highly sophisticated devices, significantly increasing their utility in recent years. These materials must fulfill several crucial requirements, such as delivering oxygen to tissues, managing refractive changes, safeguarding tissues during surgery, facilitating tissue integration, and modulating the healing process [111,112]. The recent advancements in biomaterials for treating DED include scaffolds, nanosystems, hydrogels, and drug-eluting contact lenses (Figure 4a). They are described below.

### 6.1. Scaffolds

Scaffolds have a crucial function in ex vivo tissue engineering methods for various organs, offering numerous advantages in the creation and transplantation of organs. First, they enable a greater quantity of viable cells to be transplanted, which is vital for fully restoring organ function. Scaffolds offer an ideal framework for diverse cells to coalesce and thrive within a controlled microenvironment. Second, scaffolds aid the process by offering surfaces where different cell types can flourish, directing their growth to precise locations for functional outcomes. Additionally, the meticulously selected physical and chemical attributes of biomaterials, including factors like strength and degradation rates, can be customized to boost specific functions of the emerging tissues, such as in the context of the lacrimal gland.

The introduction of organ-on-a-chip technology brought about the use of three-dimensional (3D) methods, incorporating microfluidics and bioengineering, to replicate in vivo conditions [113]. An instance of this technology in the field of ophthalmology is the creation of a human blinking eye-on-a-chip [114]. In this model, 3D shell scaffolds are utilized to create corneal curvatures. These scaffolds are infused with primary human keratocytes and placed between a microfluid channel and a circular chamber. Epithelial cells are then strategically positioned on the scaffold using a color-coded method, employing green fluorescence in the center and red fluorescence along the periphery of the scaffold surface. Additionally, 3D-printed eyelids, designed to simulate natural blinking, are mechanically activated, enabling the replication of tear-film spreading and hydration of the ocular surface.

The creation of scaffolds through different methods, the use of appropriate biomaterials, and thorough biological evaluations of relevant parameters are viable options. Both 2D and 3D culturing techniques continue to be valuable for assessing various in vitro and in vivo cultures, considering functional parameters. This approach can lead to the development of an effective ex vivo manufacturing process and enable post-implantation assessments, potentially eliminating ocular surface disorders associated with DES [115].

### 6.2. Nanosystems

Nanosystems have been extensively studied in medicine, including their use in treating eye conditions [116]. Their complex, nanoscale structure shows significant promise in improving ocular drug delivery through the controlled release of different bioactive substances. Furthermore, these nanosystems have an enhanced ability to infiltrate and pass through ocular tissues, simultaneously protecting bioactive molecules from degradation [117]. A key benefit of employing nanosystems in delivering drugs to the eyes lies in their mucoadhesive properties. This characteristic boosts their ability to stick to the ocular surface, preventing the drugs from being washed away by the eye’s natural defense mechanisms [118].

Different types of nanosystems, including nanoparticles (NPs), nanoemulsions, lipid nanocapsules, and nanoemulsions, have been investigated for transporting drugs such as epigallocatechin gallate (EGCG), cyclosporine, dexamethasone, amfenac, and cyporin-N to the eye’s surface [119]. EGCG, an anti-inflammatory substance, was integrated into gelatin nanoparticles that are biocompatible and biodegradable, and these were further coated with HA [120]. The EGCG nanoparticle formulation exhibited enhanced penetration into human corneal epithelial cell cultures. When administered twice daily over a two-week period, these nanoparticle eye drops not only reinstated tear production but also repaired the impaired corneal epithelium in a rabbit model of DED. These effects surpassed those observed with standard EGCG eye drops. Additionally, a multifunctional therapeutic gold nanoparticle was developed, featuring a substantial surface area and combining anti-inflammatory (amfenac) and antioxidant (catechin) agents to address DED [121]. Poly(catechin) capped gold nanoparticles were designed to include the anti-inflammatory drug amfenac, effectively suppressing both DED-related inflammation and reactive oxygen species (ROS)-mediated processes within four days in the rabbit DED model induced by BAC. These nanoparticles have a loose polymeric matrix containing the drug, which is uniformly confined on the gold nanoparticles’ surface. This outperformed the effects of commercial cyclosporine eye drops.

In 2016, Liu and his team investigated an innovative mucoadhesive nanoparticle system. This system involved poly(D,L-lactide)-b-dextran (PLA-b-Dex) particles coated with phenylboronic acid (PBA) to prolong the retention of eye drops. This system was investigated with the inclusion of cyclosporine. Lipid nanocapsules (LNCs) are utilized for lipophilic drugs, with the lipid core providing enhanced nano-encapsulation of the drug. LNC eye drops containing CsA demonstrated quicker and more effective therapeutic outcomes in a rat model of DED, with improved TBUT (>8 s), a decreased corneal fluorescein score, and low expression of inflammatory cytokines, surpassing the effects of the commercial CsA emulsion (Restasis) [122]. Nanowafer (NW) is a drug delivery nanosystem consisting of small drug-containing nanoreservoirs arranged on a circular transparent disc [123].

Restasis, a 0.05% CsA emulsion, was the first CsA formulation approved by the Food and Drug Administration (FDA) for the treatment of DED in 2003 [124]. Safety assessments in Phase III trials revealed that Restasis was associated with sensations of burning, foreign body presence, and stinging in 25% of patients. These effects were attributed to the use of a high total drug dosage [125]. In 2003, TJ Cyporin, a 0.05% cyclosporine A nanoemulsion, received approval for treating DED in South Korea. A study conducted by Kang and colleagues compared its effectiveness with Restasis in patients with primary Sjögren syndrome. Cyporin-N exhibited a significant improvement in TBUT after 12 weeks, while Restasis did not show the same improvement. Both treatments effectively reduced inflammation in Sjögren’s syndrome patients, with no notable difference in the reduction of inflammatory cytokines between the two groups. Additionally, in 2018, the US FDA approved Cequa^®^ (manufactured by Sun Pharma, Mumbai, India), a preservative-free 0.09% nanomicellar formulation of cyclosporine A, for the treatment of DED in adult patients [126]. Nanosized hydrogels, known as nanogels, have been widely utilized in ophthalmic applications, mainly due to their prolonged ocular retention and low viscosity [127]. In a rabbit model of DED, administering nanogel eye drops twice daily provided faster and more efficient relief for dry eyes compared to the commercially available highly viscous Vidisic gel containing 0.2% poly(acrylic) acid. Dendrimers, a unique category of nanosystems, are complex, branched molecules with diverse functional groups and intricate polymeric structures. Catechins are known for their anti-inflammatory, antibacterial, and anticancer properties. A nanocomplex consisting of PEG and catechin significantly increased tear production in a mouse model of DED while also reducing fluorescein and corneal irregularity scores [128]. Table 3 summarizes the outcomes from various studies exploring nanosystems for treating DED.

### 6.3. Hydrogels

Hydrogels, made of absorbent hydrophilic polymers, maintain their 3D structure while absorbing water. They are formed from natural, semisynthetic, or synthetic polymers like HA and poly (acrylic acid). Hydrogels offer controlled drug release, biocompatibility, and the ability to carry diverse drugs, making them promising for ocular surface disease treatment [131].

Several hydrogel products for DED treatment are available in the market, such as Hylo^®^gel (URSAPHARM, Saarbrücken, Germany), Vidisic^®^ gel (Bausch and Lomb, Rochester, NY, USA), GelTears^®^ (Bausch and Lomb, Rochester), Viscotears^®^ (Novartis, Basel, Switzerland), and Clinitas gel^®^ (Altacor, Reading, UK). Some products are in clinical trials, including VisuXL^®^ gel (VISUfarmaSpA, Rome, Italy), and bovine basic fibroblast growth factor (bFGF) gel (Zhuhai Yisheng Biological Pharmaceutical Co., Ltd., Zhuhai, China). Recent patents for DED treatment involve innovative hydrogel formulations like multi-arm PEG insert with CsA/Dex, PNIPAAm and butyl acrylate plug, guar gum, PVA, and boric acid drop containing diquafosol sodium [132,133,134].

Hydrogels containing HA have been explored in a few rabbit DED models. Soft hydrogels are well-tolerated on the ocular surfaces of rats, rabbits, and dogs [131]. In a clinical trial, canines suffering from DED and previously treated with artificial tears and cyclosporine exhibited alleviated clinical symptoms in more than 65% of instances, even if they had not responded to cyclosporine treatment initially. Thermo-responsive hydrogels have attracted considerable attention in the realm of hydrogel-based drug delivery systems due to their ability to change their physical form in response to external factors such as temperature, pH, and ionic strength [135]. Another study utilized crosslinked modified HA to create a hydrogel with higher viscosity and elastic modulus compared to non-crosslinked HA solutions. This hydrogel improved TBUT in rabbits and was found to alleviate dry eye symptoms in dogs in a preliminary clinical study, outperforming commercial HA tears in terms of symptom relief [136].

Researchers have studied hydrogels as plugs for the lacrimal drainage system in models of DED. In a rabbit DED model, a thermosensitive hydroxybutyl chitosan (HBC) hydrogel plug demonstrated notable enhancements in tear volume and decreased outflow [137]. This gel was safe and well-tolerated in both animal and human evaluations. Additionally, a novel mini eye patch containing palladium-coated gold nanorod hydrogel was developed to stimulate lacrimal tear secretion using visible light energy. The eye patch was proven safe and effective in improving tear-related parameters in healthy volunteers [138]. However, its impact on DED patients remains to be explored. Table 4 summarizes the outcomes from different studies exploring hydrogels for treating DED.

### 6.4. Drug-Eluting Contact Lenses

Delivering drugs using contact lenses (CL) has numerous benefits, including prolonged drug release, enhanced drug absorption, improved patient adherence, and greater comfort [142]. Advancements in CL materials and better patient education have resolved initial concerns, making CL more comfortable and increasing global adoption in the past decade [143]. Soft CL for delivering drugs to the eyes was first introduced in the 1960s. However, notable advancements in biomaterials from the 2000s onward have sparked increased research into CL drug delivery. Hydrogels and silicone hydrogels are the main materials employed in creating these lenses. Drug-releasing CL releases drugs that fall into two categories: lubricants and anti-inflammatory drugs. Various techniques, such as soaking, molecular imprinting, ring implantation, and incorporating nanocarriers and functional molecules, are used to load these drugs into the lenses (Figure 4b).

Researchers have investigated the use of CL as a delivery method for lubricants and anti-inflammatory drugs. Studies have successfully achieved sustained drug release over extended periods, ranging from 48 h to 15 days, in an in vitro experiment. Maulvi and colleagues (2015) investigated two methods of loading HA into hydrogel CL: soaking and direct entrapment during polymerization. Soaking allowed HA release for up to 48 h, while the entrapment method achieved a prolonged release lasting up to 264 h at therapeutic levels in vitro. These hydrogels were non-toxic and, with the direct entrapment method, demonstrated prolonged HA release for 15 days in the precorneal area of rabbits. Additionally, ring implants not only prolonged the presence of HA and maintained its continuous release onto the eye surface but also decreased corneal epithelial damage and reinstated tear volume in a BAC-DED rabbit model when compared to untreated eyes [144]. To enhance the sustained release, chitosan nanoparticles loaded with HA were integrated into ring-shaped poly(vinyl) alcohol hydrogels that were implanted in CL and that were able to release HA for 14 days [145].

In a recent study, a porous carrier loaded with CsA was created utilizing a supercritical fluid method and incorporated into a hydroxyethyl methacrylate (HEMA) hydrogel CL. This resulted in an initial rise in CsA concentration, followed by sustained release lasting 48 h. Its application resulted in elevated tear volume and extended TBUT, along with decreased corneal staining scores in the rabbit DED model in contrast to 0.05% CsA eye drops, balanced salt solution (BSS), and soft contact lenses. Additionally, CsA-CL significantly lowered pro-inflammatory cytokine IL-1β levels compared to BSS-soft CL groups [146]. The inclusion of vitamin E in CL prolonged the dexamethasone release significantly [147]. When loaded with 30% vitamin E, the release time was extended to 7–9 days. For p-HEMA hydrogel lenses, CsA was released for one day, while for silicone hydrogel lenses, the release continued for two weeks. [148]. Table 5 summarizes the results from different studies using various drug-eluting contact lenses for treating DED, and Table 6 summarizes the pros and cons of the most commonly used contact lenses for treating DED.

### 6.5. Contact Lens-Based Drug Delivery for DED

The objective of managing dry eye is to alleviate symptoms, enhance the eye’s surface, enhance vision, and tackle root causes. Various therapies, including drug delivery through contact lenses, have been proposed to manage the complex nature of DED. CsA can be incorporated into contact lenses for treating dry eyes. Compared to corticosteroids, CsA offers advantages such as reversible effects, minimal systemic absorption, and no notable side effects. These pharmacokinetic benefits are crucial for the prolonged treatment of chronic conditions like dry eye [153].

In a study conducted by Mun and colleagues in 2019, they showed that the liberation of CsA from CL significantly enhanced the treatment of DED in rabbit eyes. To induce DED, they administered 3-concanavalin A injections to rabbits and discovered that the contact lens facilitated consistent CsA release for a duration of 7 days. The effectiveness of the treatment was confirmed through corneal immunofluorescence staining, focusing on MMP9, a marker for DED [154]. In a study conducted by Desai and colleagues in 2022, they observed a decrease in MMP9 intensity in the right eyes treated with CsA/C-HA micelle contact lenses and eye drops compared to the control group in the left eyes. Moreover, rabbits exhibited swift recovery from DED when using a graphene contact lens loaded with CsA, ensuring a continuous high concentration of CsA in the corneal fluid [155].

### 6.6. Biosensors Integrated Contact Lenses

A biosensor is a diagnostic tool that utilizes a biological element and a physicochemical indicator to detect a specific chemical substance [156,157]. It employs biological components like enzymes, antibodies, cell receptors, etc. to interact with the target analyte. The biosensor works through various mechanisms, such as piezoelectric, electrochemiluminescence, optical, and electrochemical, to measure and quantify the analyte. The results are displayed simply through a connected reader [71]. The human eye holds essential chemical, physical, and biological information relevant to human health. This aspect has become a focal point for the advancement of soft electronic systems employed in diagnosing different eye ailments and other organ-related disorders. Wearable and pliable medical devices such as CL can take on pivotal tasks in the diagnosis and management of ocular diseases [158].

**Table 6 bioengineering-11-00039-t006:** Advantages and disadvantages of most commonly used contact lens biomaterials.

Contact Lens Biomaterials	Pros	Cons	References
Poly (vinyl alcohol) (PVA)	- Cost effective - Biocompatible - Effortless manufacturing	- Less permeability to oxygen - Fixed water contact	[159]
Silicon hydrogel	- High permeability to oxygen - High strength and longevity	- Costly - Aggressive conduct	[160]
HEMA hydrogel	- Cost effective - Biocompatible - Various copolymer options	- Low permeability to oxygen - Protein deposition issues	[161,162]
Polymethyl methacrylate (PMMA)	- Low cost - Extensively studied polymer	- Impermeable to oxygen - Rigid in eyes - Hostile conduct	[161,163]

## 7. Challenges and Future Perspectives

Current treatments for ocular diseases are mainly palliative, relying on lubricating agents like artificial tears and gels. Advanced therapies like anti-inflammatory therapy with drugs or AS offer some relief but do not eliminate the disease. Cell-based approaches, such as salivary gland transplantation, show promise but have limitations. More research is needed to find a long-term solution. Medical advancements focus on regenerative approaches and tissue engineering for ocular surface regeneration. This involves studying stem cells and their markers to trigger tissue recovery, such as in lacrimal glands. Regenerative remedies include pharmacological drugs (cyclosporine A, corticosteroids, etc.) and treatments like autologous PRP injection and tumor necrosis factor-stimulated gene-6 (TSG-6) [39,164]. They are being investigated to assess their influence on the clinical symptoms of DES. Promising regenerative approaches in medicine involve MSCs, which have shown anti-inflammatory properties and tissue repair potential. Studies have demonstrated MSCs can reduce inflammatory damage and promote growth and repair through the release of growth factors [105]. Evidence from animal models and in vitro studies supports their potential for treating DES. Similarly, human amniotic membrane epithelial cells and iPSCs show regenerative potential in research, but clinical reports in humans are lacking. GMP-compliant culturing protocols are crucial for the large-scale clinical use of these cells. Gene therapy is another area of exploration, with potential targets identified for manipulating gene expression using specific molecules. While progress has been made in animal studies, clinical experiments are needed. Overall, advancements in treating DED show promising outcomes; however, further clinical studies are essential to advance these approaches.

Advancements in biomaterials technology have opened up promising avenues in dry eye research [159]. One crucial aspect is obtaining regulatory approval for these new devices, mainly because they fall under the category of medical devices. While animal models have been extensively used to study their safety and ocular toxicity, there is a need for more investigations into their effects on human tissues. Although these biomaterials offer advantages over conventional administration methods, further enhancements are necessary to improve their efficacy and design [165]. A breakthrough in biomaterials technology for research and development has opened up a promising avenue in the field of DED. This approach offers significant advantages over traditional delivery methods. However, there is still room for enhancing the design and effectiveness of these biomaterials. Exploring the impact of the composition, particle size, and surface charge of nanosystems on pharmacokinetics is an important aspect of this research [119].

To advance this field, standardization efforts should be intensified to refine large-scale manufacturing. Additionally, conducting extensive and long-term studies, including clinical trials, is vital to assessing ocular toxicity and biocompatibility. This process is essential for smoothly transitioning these therapies into clinical use. Similarly, there is a need for deeper exploration to understand the interactions between drugs and hydrogels, as well as drug-releasing contact lenses, within the bodies of individuals with DED. For instance, it is essential to perform thorough examinations of nanosystems’ particle size, composition, and surface charge in pharmacokinetic research [119,166].

Moreover, it is vital to explore research on hydrogels’ roles as ocular drug delivery systems, ensuring the maintenance of their 3D structures and the effectiveness of the loaded drugs. Addressing concerns related to CL, especially discomfort following their prolonged use and the risk of infections and systemic toxicities, is crucial. Some studies have explored the possibility of preventing these issues by applying antibacterial coatings to CL. Moreover, comprehensive in vivo tests with drug-releasing CL are essential to validate their effectiveness in delivering drugs to individuals with DED. Additionally, conducting extensive large-scale production, late-phase clinical trials, and extended investigations into ocular toxicity and biocompatibility are imperative for validation purposes [167]. To introduce this innovative ocular drug delivery system to the market, it is crucial to undertake thorough studies that bridge the existing gaps in knowledge and technology [33].

## 8. Conclusions

As the population ages, the prevalence of eye diseases is rising. Consequently, biomaterials, tissue engineering, and regenerative medicine have gained increasing importance in the field of ophthalmology by restoring eyesight and enhancing the quality of life for numerous patients. The innovative therapies highlighted in this review exhibit great promise in mitigating the impact of DED and offer distinct advantages over the traditional use of lubricant eye drops, which provide only temporary relief. Moreover, these new therapies present a solution for managing the advanced stages of the disease by addressing its root cause. However, it is crucial to acknowledge that, while these innovative therapies exhibit great potential, many are still in the theoretical or animal-based stages of development, with limited human trials in some cases. This underscores the need for continued research and clinical investigations to validate their safety, efficacy, and applicability in human populations. Future endeavors should focus on bridging the gap between preclinical success and successful translation to human trials, ensuring that these promising therapies fulfill their potential in benefiting patients and improving overall visual health.

## Figures and Tables

**Figure 1 bioengineering-11-00039-f001:**
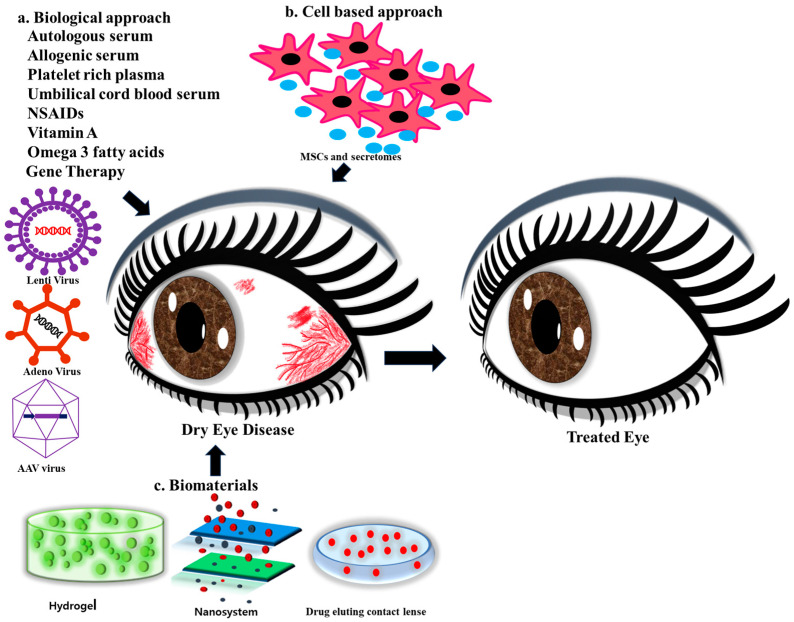
Different regenerative approaches to treating DED (a) Biological approaches (gene therapy, autologous serum, allogenic serum, platelet-rich plasma, umbilical cord blood serum, NSAIDs, Vitamin A, Omega 3 fatty acids). (b) Cell-based therapy, (c) Biomaterials for treating DED.

**Figure 3 bioengineering-11-00039-f003:**
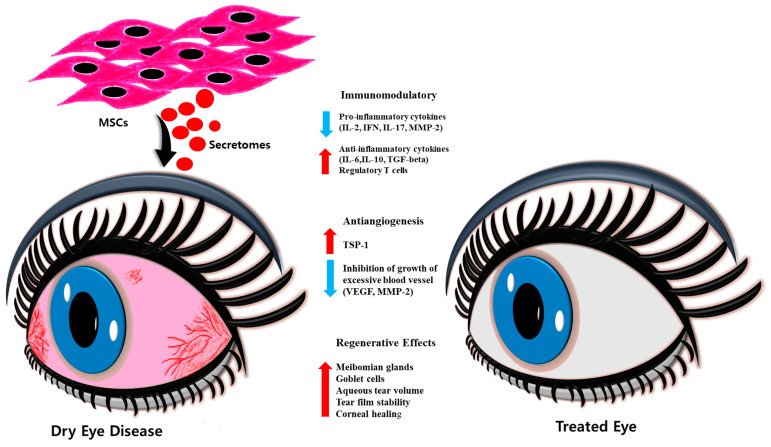
Cell-based approaches in treating DED. The cells derived from different sources perform immunomodulatory activities by liberating pro-inflammatory and anti-inflammatory cytokines and help in treating DED.

**Figure 4 bioengineering-11-00039-f004:**
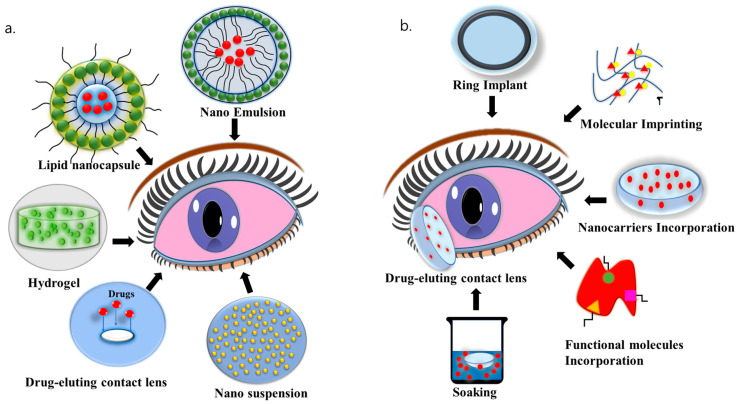
(**a**) Different types of biomaterials like nano systems (nanosuspension, nano emulsion, lipid nanocapsule), hydrogels, and drug-eluting contact lenses for treating DED. (**b**) Several methods for integrating drugs into contact lenses.

**Table 1 bioengineering-11-00039-t001:** Clinical trials using various biological approaches in treating DED.

Approaches	Population (n)	Treatment Period	Results	References
Autologous serum	48	6–10/day (6 months)	- Improvements in symtoms - Increase corneal sensitivity - Increase in goblet cell density	[62]
Autologous serum	144	6/day (6 weeks)	- Notable enhancements in clinical parameters and tear protein profile (lysozyme) - Sustainable increment in total tear protein level	[84]
Autologous serum	240	4/day (12 weeks)	- Impovement in ocular surface diseases index (OSDI) score, TBUT, Schimer I test, corneal flurescein staining, and conjuctival imoression cytology	[46]
Autologous serum	27	5/day (6 months)	- Improvements in TBUT and fluorescein scores	[85]
Platelet-Rich Plasma	30	5/0, 30, 60, and 90 days	- Increase in the mean Schirmer value - Increase in TBUT - Improvement in subjective OSDI values - Decrease in corneal staining	[86]
Platelet-Rich Plasma	47	4/day (3 months)	- Improvements in clinical symtoms - Decrease in corneal staining	[87]
Platelet-Rich Plasma	368	6/day (6 weeks)	- Decrease in OSDI and the Oxford scale of corneal fluorescein staining	[59]
Platelet-Rich Plasma	3	6/day (3 months)	- Improvements in the symptoms and visual acuity - Improvement in corneal sensitivity - Decrease in fluorescein corneal staining	[88]
Platelet-Rich Plasma	360	6/day (6 weeks)	- Imrovement in the clinical symtoms - Decrease in the ocular surface diseases index and the Oxford scale of corneal fluorescein staining	[59]
Allogenic serum	49	6/day (1 month)	- Improvements in mean OSDI score and TBUT	[89]
Allogenic serum	16	6–8/day	- Decrease in OSDI, tear osmolarit, corneal staining scores - Improvements in the goblet cell density, TBUT and Schirmer I test	[90]
Umbilical cord blood derum	20	8/day (2 months)	- Significantly improved in corneal epitheliopathy - Decrease in Oxford staining score	[91]
Umbilical cord blood derum	60	8/day (1 month)	- Reduction in corneal staining - Decrease in visual analogue score (VAS) and OSDI - Downregulation of IL-13	[92]

**Table 2 bioengineering-11-00039-t002:** Pre-clinical and clinical studies using cell-based approaches in treating DED.

Type of Cells	Human/Animal	Number	Mode of Treatment	Results	References
Allogeneic bone marrow-derived MSCs (BM-MSCs)	Mice	20	Intraperitoneal injection	- Increase in tear production - Decrease in inflammatory cytokines	[105]
Human MSCs	Mice	N/A	Periorbital injection	- Preservation of corneal epithelial integrity - Suppression of the inflammation - Increase in tear production - Increase in conjuctival goblet cells	[106]
Mesenchymal stromal cells-derived extracellular vesicles (MSC-EVS)	Mice	6	Topical drop	- Suppression of ocular surface inflammation by inhibiting dentritic cells - Downregulation of inflammatory cytokines (TNF-α, IL-6, and IL-1β)	[107]
Allogenic adipose-derived bone marrow MSCs (AD BM-MSCs)	Rats	16	Tropical drop	- Increment in the number of secretory granules and goblet cells - Increase in aqueous tear volume	[96]
Umbilical cord-derived -MSCs (UC-MSCs)	Rabbits	36	Intravenous injection	- Suppression of the inflammation (TNF-α, IL-1β, and IL-6)	[108]
Allogenic adipose derived-MSCs (AD-MSCs)	Mice	18	Tropical drop	- Inhibition of cell apoptosis - Downregulation of anti-inflammatory cytokines (IL-1β, IL-6, IF-γ, and IL-18)	[101]
BM-MSCs	Humans	20	Intravenously	- Suppression of the inflammatory and fibrous processes in eyes	[109]
AD-MSCs	Humans	7	Transconjuctivital injection	- Improvement of both signs and symptoms with single injection of allogenic ASCs - Increment in TBUT and Schirmer’ I test - Decrease in tear osmolarity	[98]
AD-MSCs	Humans	61	Transconjuctivital injection	- Increase tear film stability - Decrease ocular discomfort	[110]

**Table 3 bioengineering-11-00039-t003:** Different Nano-systems for Treating DED.

Nano-Systems	Method of Inducing DED	Animals	Treatment Period	Outcomes	References
Gelatin nanoparticle	0.1% BAC	Rabbits	21 days (twice daily)	- Downregulation of TNF-α, IL-8, and IL-6 - Decrease in fluorescein score - Increase in tear secretion	[120]
Gold/poly(catechin) core-shell nanoparticle	0.15% BAC	Rabbits	4 days	- Decrease in fluorescein score - Decrease in ROS - Decrease in inflammation	[121]
Glycol chitosan nanoparticle	Subcutaneous injection of scopolamine hydrobromide in mice	Mice	7 days	- Decrease in ROS - Increase in tear production - Promotion of corneal and conjuctival cell growth and integrity	[129]
Cationized gelatin and chondroitin sulfate nanoparticles	Subcutaneous injection of scopolamine + desiccating stress	Mice	5 days	- Increase in tear production - Upregulation of goblet cells - Reduction in the CD4+ T-cells infiltration in the conjuctiva	[130]

**Table 4 bioengineering-11-00039-t004:** Different Hydrogels for treating DED.

Hydrogel	Method of Inducing DED	Animals	Treatment Period	Results	References
Crosslinked thiolated carboxymethyl HA	Diagnosed with DED	Dogs	2 times each day for 2 weeks	- Decreased DED symptoms in dogs - Improvement in ocular irritation, conjunctival hyperemia, and ocular discharge	[139]
Crosslinked thiolated carboxymethyl HA	Diagnosed with DED	Dogs	3 weeks 3 times daily	- Decreased DED symptoms in dogs - Significantly improved ocular surface health (ocular iritation, and ocular discharge)	[136]
Gelatin, poly(N-isopropylacrylamide), lectin helix pomatia agglutinin and EGCG drug	0.1% BAC twice daily for 14 days in rabbits	Rabbits	One-time adminis-tration	- Decreased ROS level - Decreased MCP-1 levels - Downregulation of inflammation	[140]
FK506 loadedMPEP hydrogel	Scopolamine mice model	Mice	5 days, twice daily	- Decreased in MMP-13 levels - Decreased inflammation - Increased goblet cells - Increased tear production	[141]
Hydroxybutyl chitosan as intracanalicular injection	0.1% BAC for 5 weeks in rabbits	Rabbits	One-time intracan-alicular injection	- Increased tear production - Decreased inflammation	[137]

**Table 5 bioengineering-11-00039-t005:** Different drug-eluting contact lenses for treating DED.

Contact Lens Material	Drug/Molecule	Method of Drug Loading	Animals	Treatment Period	Outcomes	References
HEMA hydrogel	HA	- Soaking and direct entrapment	Rabbits	15 days	- Transmittance at 95% with soaked HA CL - Decrease in transmittance with HA entrapped CL - Increase in HA mean residence time and area in pharmacokinetics studies in rabbit tear fluid	[144]
HEMA hydrogel	HA	- Ring implant and soaking	R RRabbits	15 days	- Decreased contact angle with increase in HA loading - Increment of residence time of HA - Faster and complete healing of DES	[149]
HEMA hydrogel	PVP-K90	- Soaking - Direct entrapment	Rabbits	-	- Downregulating corneal epithelium damage - Upregulation of tear volume - Transmittance more than 95%	[150]
Silicone hydrogel	Flurbiprofen sodium diclofenac sodium ketorolactromethamine	- Vitamin E loading + cationic surfactants	-	-	- Enhanced both the loading and prolonged release of medications within the therapeutic period	[151]
Silicone hydrogel	Pirfenidone	- 20% vitamin E loading	Rabbits	3 h	- Drug presence in the aqueous humor for 3 h - Decrease in expression of TNF-α, IL-1β, and TGF-β1	[152]

## Data Availability

Not applicable.

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
