# Peer review of "Current Advances in Regenerative Strategies for Dry Eye Diseases: A Comprehensive Review"

_bioengineering, 2023, doi:10.3390/bioengineering11010039_

Round 1
Reviewer 1 Report
Comments and Suggestions for Authors
In this review article, the authors discussed advancements in the treatment of dry eyes. The dry eyes is one of the serious issue in ophthalmic diseases so this discussion and future directions would be useful for the researchers working on ophthalmic diseases. So, it can be accepted for publication.
Author Response
Dear Editor-in-chief,
We appreciate your sending the sincere feedback quickly and the comments that reviewers found interesting to our manuscript “Current Advances in Regenerative Strategies for Dry Eye Diseases: A Comprehensive Review” (bioengineering-2749943). We made corrections focusing on the comments from reviewers. In addition to this, we checked the entire contents of the manuscript and corrected the sentences. Many edits have been made to reduce the ambiguity of the overall content flow and make it easier for readers to understand. A list of changes is highlighted with red in the revised manuscript.
Response to Reviewer:
Reviewer 1 comments
#1 In this review article, the authors discussed advancements in the treatment of dry eyes. The dry eyes is one of the serious issue in ophthalmic diseases so this discussion and future directions would be useful for the researchers working on ophthalmic diseases. So, it can be accepted for publication.
Response #1
I am grateful for your thorough review of our manuscript. We appreciate the time and effort you invested in evaluating our work. We are delighted to learn that you find our discussion on advancements in the treatment of dry eyes to be of significance in the context of ophthalmic diseases. We sincerely express our gratitude for your recommendation in acceptance for publication.
I gratitude for the reviewer’s advice.
With best regards,
Jae Yong Kim, MD, PhD
Professor
Department of Ophthalmology, Asan Medical Center,
University of Ulsan College of Medicine, Seoul 05505, Republic of Korea
Email: jykim2311@amc.seoul.kr

Reviewer 2 Report
Comments and Suggestions for Authors
The manuscript summarized well about the regenerative strategies for DED and organized with separated sections for DED treatment.
However, if authors organized with patients data and animal experiments, it would be more structurally and logically easy to follow the idea of applications. In addition, in human trial or approved application table insertion might be improved the manuscript quality.
Although, authors categorized with biological approach, material differences, and others, major focus was utilizing CsA or HA application with each sections. As DED/DES is "a mulitifactorial condition" involved with eye surface, there would be more applications or trials need to described.
In some trials, NSAIDs are applied to the DED/DES models, multi-vitamins or dietary supplements are treated to the patients or models, and many other treatments are described in other aspects of studies.
Therefore, authors should consider those other treatments or applications in the manuscript for DED/DES.
In the manuscript, there are many factors mentioned for the reason of outbreak of DED. It would be complicated to draw a one summary mechanism of action figure, but be helpful to the readers understanding how those factors (cytokines, cell types, tissues, etc) are connected.
As a minor comments, please check the typo on the manuscript. Line 293 Tfh seems to be typo and in figure 3, closing parenthesis is missing after MMP-2.
Comments on the Quality of English LanguagePlease, check the manuscript with any typo carefully.
Author Response
Dear Editor-in-chief,
We appreciate your sending the sincere feedback quickly and the comments that reviewers found interesting to our manuscript “Current Advances in Regenerative Strategies for Dry Eye Diseases: A Comprehensive Review” (bioengineering-2749943). We made corrections focusing on the comments from reviewers. In addition to this, we checked the entire contents of the manuscript and corrected the sentences. Many edits have been made to reduce the ambiguity of the overall content flow and make it easier for readers to understand. A list of changes is highlighted with red in the revised manuscript.
Response to Reviewer:
Reviewer 2 comments
#1 The manuscript summarized well about the regenerative strategies for DED and organized with separated sections for DED treatment.
However, if authors organized with patients’ data and animal experiments, it would be more structurally and logically easy to follow the idea of applications. In addition, in human trial or approved application table insertion might be improved the manuscript quality.
Response #1
We appreciate your insightful and constructive feedback on our manuscript. We sincerely appreciate the time and effort you devoted to the review process.
We are pleased to hear that you found the manuscript's summary of regenerative strategies for DED to be comprehensive and well-organized. Your suggestion regarding the organization of patient data and animal experiments is valuable, and we acknowledge the merit in enhancing the structural and logical flow of our work. Thus, we have added the preclinical and clinical trials in our manuscript in table and text form (Table-1 and Table-2), section 4.1- (edit lines: 189-193, and 197-199), section 4.3-(edit lines: 251-253), section 4.4 (edit lines: 264-266), section 4.5- (edit lines: 301-306), section 4.9-(line numbers:351-356), and section 4.11-(edit lines: 376-379,381-383).
#2 Although, authors categorized with biological approach, material differences, and others, major focus was utilizing CsA or HA application with each sections. As DED/DES is "a mulitifactorial condition" involved with eye surface, there would be more applications or trials need to described.
In some trials, NSAIDs are applied to the DED/DES models, multi-vitamins or dietary supplements are treated to the patients or models, and many other treatments are described in other aspects of studies.
Therefore, authors should consider those other treatments or applications in the manuscript for DED/DES.
Response #2
We are very grateful for the reviewer’s insightful comments on our paper. We have mentioned reviewers’ opinion in the edited lines: 318-324 and 540-553.
Furthermore, we have expanded the coverage of treatment strategies within the biological approaches’ sections, incorporating additional information on NSAIDs, punctal plugs, corticosteroids, Vitamin A, and Omega-3 fatty acids. These additions care mentioned in Sections 4.7 to 4.11, aligning with the constructive feedback provided by the reviewer.
#3. In the manuscript, there are many factors mentioned for the reason of outbreak of DED. It would be complicated to draw a one summary mechanism of action figure, but be helpful to the readers understanding how those factors (cytokines, cell types, tissues, etc) are connected.
Response #3
Thank you for your thoughtful evaluation of our manuscript. We appreciate the time and consideration you invested in providing valuable feedback. We have added one more figure showing the mechanism of DED involving various cytokines, cell types (Figure 2B). Further we have explained this in the section 2.2 (edit lines: 102-124), Figure legend 2B (edit lines-145-150)
#4. As a minor comments, please check the typo on the manuscript. Line 293 Tfh seems to be typo and in figure 3, closing parenthesis is missing after MMP-2.
Response #4
We are very grateful for the reviewer’s insightful suggestions. We have made the correction in the typo (Edit Line - 409) and figure 3 parenthesis after MMP-2
#5 Comments on the Quality of English Language
Please, check the manuscript with any typo carefully.
Response #5
Thank you for your careful review of our manuscript and for bringing attention to the aspect of the English language quality, particularly with regard to potential typos. We appreciate your diligence in ensuring the overall excellence of the manuscript.
In response to your suggestion, we have thoroughly reviewed the manuscript to identify and correct any typos or language-related issues.
I gratitude for the reviewer’s advice.
With best regards,
Jae Yong Kim, MD, PhD
Professor
Department of Ophthalmology, Asan Medical Center,
University of Ulsan College of Medicine, Seoul 05505, Republic of Korea
Email: jykim2311@amc.seoul.kr

Reviewer 3 Report
Comments and Suggestions for Authors
Nicely written. Very well thought out. Comments are in two varieties.
1: Simple ones
lines 66-69: the 1st 'and' should be a comma?
Figure 1: "Vicious cycle and inflammation" part not explained anywhere? Why is it there?
line 95: insert "(figure 2)" (remove from line 98)after "hyperosmolarity" as that is that the figure is about.
line 136: "...EITHER (delete "swiftly") enter the nasolacrimal duct OR (delete but") are..."
2: More detailed issues: about organization
Sections 4 - 6: Written inconsistently. Sometimes there is a lot of text (with experimental explanations: 4 and 5), sometimes there are incomplete tables with repetitive experimental explanations (6.2/table 1, 6.4/table 3) and sometimes there is an explanation of the method, with less experimental details with a table with experimental details ((6.3/table 2). The function of tables is to remove detail orientated text from the main body, and to replace that with overall explanations of what is where and what does it all mean. 6.3/table 2 is the most successful at doing that. Thereby the busy clinician can rapidly read through the text to see what they need to know, and use the table to find the specific details.
With this in mind, section 7 needs to be redistributed to each of the previous sections ( 4-6) to better emphasize what is known and what needs as yet to be done. Currently it reads like a bunch of unrelated paragraphs. Also, under-referenced.
Also, please make sure to clearly note delivery mechanisms for each section (e.g.: section 4 is all topical?)
Table 4 is interesting, but not cited in text.
Conclusion is weak - after all that detail, it is underwritten. This is an opportune time to clearly state that many of the newer technologies are theoretical or animal based only. That human trials are only present in some cases.
Comments on the Quality of English Language
Some minor oddities noted in the previous section.
Author Response
Dear Editor-in-chief,
We appreciate your sending the sincere feedback quickly and the comments that reviewers found interesting to our manuscript “Current Advances in Regenerative Strategies for Dry Eye Diseases: A Comprehensive Review” (bioengineering-2749943). We made corrections focusing on the comments from reviewers. In addition to this, we checked the entire contents of the manuscript and corrected the sentences. Many edits have been made to reduce the ambiguity of the overall content flow and make it easier for readers to understand. A list of changes is highlighted with red in the revised manuscript.
Response to Reviewer:
Reviewer 3 comments
#1: Simple ones
lines 66-69: the 1st 'and' should be a comma?
Figure 1: "Vicious cycle and inflammation" part not explained anywhere? Why is it there?
line 95: insert "(figure 2)" (remove from line 98)after "hyperosmolarity" as that is that the figure is about.
line 136: "...EITHER (delete "swiftly") enter the nasolacrimal duct OR (delete but") are..."
Response #1
Thank you for your detailed feedback on our manuscript. We appreciate your thorough examination, and your insights are valuable for improving the clarity of our work.
lines 66-69: We have arranged the sentence. (Edit line:65)
Figure 1: We have removed the term Vicious cycle and inflammation from the figure.
line 95: We have arranged it according to the reviewer’s suggestion. (Edit line:97)
line 136: We have rearranged the sentence structure according to the reviewers’ suggestions. (Edit lines:158-160)
#2: More detailed issues: about organization
Sections 4 - 6: Written inconsistently. Sometimes there is a lot of text (with experimental explanations: 4 and 5), sometimes there are incomplete tables with repetitive experimental explanations (6.2/table 1, 6.4/table 3) and sometimes there is an explanation of the method, with less experimental details with a table with experimental details ((6.3/table 2). The function of tables is to remove detail orientated text from the main body, and to replace that with overall explanations of what is where and what does it all mean. 6.3/table 2 is the most successful at doing that. Thereby the busy clinician can rapidly read through the text to see what they need to know, and use the table to find the specific details.
With this in mind, section 7 needs to be redistributed to each of the previous sections (4-6) to better emphasize what is known and what needs as yet to be done. Currently it reads like a bunch of unrelated paragraphs. Also, under-referenced.
Also, please make sure to clearly note delivery mechanisms for each section (e.g.: section 4 is all topical?)
Table 4 is interesting, but not cited in text.
Conclusion is weak - after all that detail, it is underwritten. This is an opportune time to clearly state that many of the newer technologies are theoretical or animal based only. That human trials are only present in some cases.
Response #2
We sincerely appreciate your thorough and insightful evaluation of our manuscript. Your comments provide valuable guidance for enhancing the coherence and clarity of our work. We have carefully considered your feedback, and we addressed each of your points in our revision:
In sections 4-6 of our revised manuscript, we have streamlined the content by removing redundant experimental details from the paragraphs in section 6.2, and 6.4. Additionally, we have introduced separate tables, specifically Table 1 in Section 4 and Table 2 in Section 5, to present this experimental data more clearly. Similarly, to enhance clarity and comprehensibility, we have expanded the explanations in relation to the outcomes presented in Tables 3 and 4.
We have redistributed some of the paragraphs in 4-6 sections (edit lines:449-452 and 496-501). Likewise, in section 7, in our revised manuscript, we have rearranged and deleted the unrelated paragraphs and emphasized on the improvements that should be done in various regenerative approaches. (edit lines: 706-707, 708-711, 713-718, 721-723,724-731, and 737-740).
Similarly, we have added the references in the section 7. (Reference numbers- 39, 164, 105, 159, 166, 119,167, 168, and 33)
In Section 4 of our revised manuscript, we have taken careful measures to elucidate the delivery mechanisms associated with the discussed treatments. While the primary mode of delivery for most interventions in Section 4 is topical, we have conscientiously highlighted that alternative delivery modes are also equally applicable, as detailed in the revised manuscript.
Section 4.1- (Edit Lines:181-182, and 200-207)
Section 4.2-(Edit lines:212 and 219-222)
Section 4.3- (Edit lines:225-229 and 233-236),
section 4.4 -(Edit line:264-266 and 269-271),
section 4.5-(Edit lines: 287-294),
section 4.6- (Edit lines :318-324),
section 4.7-(Edit line: 326)
Section 4.9-(Edit line: 347-348)
Section 6.2-(Edit lines:504-511)
Section 6.6-(Edit lines:669-674)
Following the reviewer's guidance, we have appropriately cited Table 4 in the revised manuscript. The edited table is now numbered as Table 6, and the references related to this table are numbered as 159, 160, 161, 162, and 163 in the reference list.
In response to their valuable feedback, we have made modifications to the conclusion, addressing specific concerns outlined in the edited lines 752-759.
#3. Comments on the Quality of English Language
Some minor oddities noted in the previous section.
Response #3
Thank you for your feedback on the quality of the English language in our manuscript. We appreciate your attention to detail. To ensure the clarity and professionalism of our work, we carefully addressed the minor oddities you noted in the previous section.
I gratitude for the reviewer’s advice.
With best regards,
Jae Yong Kim, MD, PhD
Professor
Department of Ophthalmology, Asan Medical Center,
University of Ulsan College of Medicine, Seoul 05505, Republic of Korea
Email: jykim2311@amc.seoul.kr
